

# Influence of leaf morphological properties on epiphytic lactic acid bacteria counts in forage crops

Dan Wu[1,*], Guicong Tang[1,*], Gaofeng Liu[1], Ting Sun[1], Jinmei Yang[2], Guojian Tang[2] and Liuxing Xu[1]

[1] College of Agronomy and Life Sciences, Zhaotong University, Zhaotong, Yunnan, China
[2] School of Biological Sciences and Technology, Liupanshui Normal University, Liupanshui, Guizhou, China
[*] These authors contributed equally to this work.

## ABSTRACT

The structural properties of leaves play a crucial role in the attachment of lactic acid bacteria (LAB) in forage corps. This study analyzed the effects of leaf morphological properties, on LAB counts in different wild forage crops. The LAB counts and morphologic features on adaxial or abaxial surfaces of leaves from twelve forage species (maize (*Zea mays*), beggarticks (*Bidens pilosa*), white goosefoot (*Chenopodium album*), common bean (*Phaseolus vulgaris*), morning glory (*Ipomoea purpurea*), perilla (*Perilla frutescens*), tomato (*Solanum lycopersicum*), chili pepper (*Capsicum*), sweet potato (*Ipomoea batatas*), peanut (*Arachis hypogaea*), potato (*Solanum tuberosum*), and mallow (*Malva verticillata*)) were investigated. White goosefoot (5.22 $\log_{10}$ CFU $g^{-1}$ FM) and beggarticks (4.83 $\log_{10}$ CFU $g^{-1}$ FM) had the highest LAB counts but shortest leaf lengths (5.06 cm and 4.97 cm, respectively), whereas maize (3.37 $\log_{10}$ CFU $g^{-1}$ FM) and sweet potato (3.38 $\log_{10}$ CFU $g^{-1}$ FM) had lower LAB counts but significantly greater leaf widths than the other crops except for mallow ($P < 0.001$). Linear regression analysis revealed that the coefficients of determination ($R^2$) between LAB counts and contact angle of the adaxial and abaxial surfaces of leaf were 0.1424 and 0.175, respectively. Therefore, the morphological features of leaves have a relatively weak influence on the LAB counts in different forage crops.

## INTRODUCTION

The demand for animal products in China has increased in recent years (*Du et al., 2018*); however, the limited supply of high-quality forage has constrained the expansion of ruminant production (*Guo & Qin, 2022*). To reduce costs, producers have started utilizing local forage resources, including wild forage, agricultural waste (*Guo et al., 2017*), and dual-purpose (food and forage) crops (*Xu et al., 2023*). However, these forages often face challenges, such as low soluble carbohydrate content, high buffering capacity, high fiber content, and alkaloid contents, thus increasing the difficulty of producing high-quality silage. However, lactic acid bacteria (LAB) attached to the surface of forage plants can help address these issues.

Corresponding authors
Guojian Tang,
tanguojian1988@163.com
Liuxing Xu, 331405719@qq.com

LAB is a type of gut probiotic that has gained widespread attention recently because of its significant effects on promoting animal health, improving production performance, and enhancing meat and milk quality (*Da Costa et al., 2019*; *Du et al., 2023*). However, despite the important role of LAB in plant health management (*Murindangabo et al., 2023*) and silage fermentation quality (high throughput sequencing technology) (*Liu et al., 2024*), the count of LAB (tablet counting method) attached to the surface of forage grass is generally low (*Chen, Dong & Zhang, 2021*; *Wu et al., 2023*). This limits the efficiency of LAB in forage fermentation and their potential as a probiotic supplement. Notably, current research on LAB has primarily focused on their metabolic capabilities and diversity in silage (*Gao et al., 2024*; *Yang et al., 2024*), whereas studies on the attachment characteristics of LAB to forage leaves are relatively insufficient. Thus, further in-depth research is needed to reveal the key factors influencing the attachment of LAB to leaves.

The physiological and structural features of forage leaves play crucial roles in the attachment of bacteria, with factors such as surface roughness (*Crawford et al., 2012*), moisture content (*Beattie, 2011*), and stomatal density (*Yang et al., 2022*) directly influencing the attachment efficiency of these bacteria. Generally, a higher moisture content provides the necessary environment for LAB survival, thereby facilitating their initial colonization. However, different LAB have different water requirements. For example, *Lactobacillus* showed a higher positive correlation ($r = 0.72$) with water activity than *Lactococcus* and *Enterococcus* (*Minervini et al., 2015*). The higher surface roughness of the leaves provides more attachment sites, thus promoting a more stable attachment of LAB through mechanical fixation (*Sousa et al., 2011*). Stomatal density is another important factor that affects bacterial attachment. Stomata provide attachment sites and regulate the leaf surface microenvironment by controlling their opening and closing, indirectly promoting bacterial growth and reproduction (*Chaudhry et al., 2021*; *Vorholt, 2012*). Previous studies have shown that leaves with high stomatal density improve hydration (*Bertolino, Caine & Gray, 2019*) and maintain an appropriate oxygen concentration (*Royer, 2001*), leading to a higher count of attached bacteria on high-density stomatal leaves than on low-density stomatal forage. In addition, a thinner epidermis can reduce the physical barrier and thus facilitate LAB attachment, whereas a thicker epidermis barely attachment (*Underwood, 2012*). Leaf thickness can also create mechanical barriers that affect bacterial attachment (*Yang et al., 2022*). In summary, the structure of forage leaves directly determines the quantity and efficiency of LAB attachment, thereby influencing the potential applications of forage in fermentation. However, limited information is available on how leaf structure affects LAB counts.

This study analyzed the effects of the structure of different forages on LAB counts and explored approaches to promoting the proliferation of LAB on the surface of forage species. These findings provide new theoretical insights and technical support for the efficient utilization of forage resources and the application of LAB for disease prevention, fermentation, and other processes.

**Table 1  Morphological features on leaf surfaces of different forage species.**

| Forage species | Maturity stages | Morphological features | |
|---|---|---|---|
| | | Venation | Leaf shape |
| Maize | Milk stage | Vertical parallel vein | Needle shaped |
| Beggarticks | Flowering stage | Pinnate vein | Pinnate leaf |
| White goosefoot | Flowering stage | Pinnate vein | Lanceolate |
| Common bean | Podding stage | Palmate vein | Pinnate leaf |
| Morning glory | Flowering stage | Palmate vein | Heart shaped |
| Perilla | Flowering stage | Pinnate vein | Oval shaped |
| Tomato | Fruiting stage | Pinnate vein | Pinnate leaf |
| Chili pepper | Fruiting stage | Pinnate vein | Pinnate leaf |
| Sweet potato | Maturation stage | Pinnate vein | Heart shaped |
| Peanut | Maturation stage | Reticular vein | Pinnate leaf |
| Potato | Maturation stage | Reticular vein | Oval shaped |
| Malva verticillata | Flowering stage | Palmate vein | Reniform |

## MATERIALS AND METHOD

### Experimental site

The experimental materials were sampled from Zhaoyang District, Zhaotong City, Yunnan Province (27°36′N, 103°74′E) in August 2024 (annual temperature and total rainfall were 22.1 °C and 184 mm, respectively). According to data from the Zhaotong Meteorological Bureau, the average annual temperature over the past 20 years was 12.3 °C, and annual total rainfall was 682 mm.

### Forage species

This study focuses on 12 forage crops with potential feed value: maize (*Zea mays*), beggarticks (*Bidens pilosa*), white goosefoot (*Chenopodium album*), common bean (*Phaseolus vulgaris*), morning glory (*Ipomoea purpurea*), perilla (*Perilla frutescens*), tomato (*Solanum lycopersicum*), chili pepper (*Capsicum*), sweet potato (*Ipomoea batatas*), peanut (*Arachis hypogaea*), potato (*Solanum tuberosum*), and mallow (*Malva verticillata*). Among these, beggarticks, white goosefoot, morning glory, perilla, and mallow are classified as wild forage plants, whereas the others are cultivated as economic crops. Typically, the stems and leaves of these economic crops, which are left after harvesting the grains, fruits, or tubers, are either burned, mulched back into the soil, or used as animal feed. When used as animal feed, producers often mix harvested wild forage with the residues of cultivated crops to produce silage or directly feed it to livestock as fresh forage. The 12 crops were at different growth stages, and pinnate veins were the predominant leaf venation. The leaf shapes varied and included six types: reniform, pinnate, oval-shaped, needle-shaped, lanceolate, and heart-shaped (Table 1).

### Sample processing

On a sunny morning, mature and healthy leaves were collected under sterile conditions from different individuals of the same crop as experimental materials, with each crop

sampled 10 times. After collection, the materials were immediately transported in ice packs and stored at 4 °C in a refrigerator for LAB count and structural analyses.

## LAB counts and structural analyses

The LAB counts were determined using the dilution plate method. Initially, 10 g of finely chopped leaves was placed in a sterile polyethylene bag, to which 90 mL of sterile distilled water was added, and the mixture was shaken at 300 rpm for 5 min on a shaker. After shaking, an appropriate volume of the solution was collected and evenly spread on de Man, Rogosa, and Sharpe agar (Guangdong HuanKai Microbial Sci. & Tech. Co., Ltd., Guangzhou, China) plates to estimate the colony-forming units (CFU) of LAB attached to the leaf surfaces. The plates were then incubated at 37 °C under anaerobic conditions (the Petri dishes were placed in self-sealing bags, and a vacuum sealer was used to evacuate the air) for 2 d. The LAB counts were expressed as $\log_{10}$ CFU $g^{-1}$ FM (fresh matter, FM), repeated three times.

Among the leaf structural parameters, the length was determined by laying the leaves flat on a horizontal workbench and measuring the straight distance from the leaf tip to the leaf base along the longitudinal axis using a Vernier caliper. The leaf width was determined by measuring the widest part perpendicular to the longitudinal axis. A video optical contact angle meter (JC2000D1, Shanghai Zhongchen Digital Technology Equipment Co., Ltd., Shanghai) was used to measure the static contact angle of the water droplets on the adaxial and abaxial surface of the same leaf. Leaf samples were sectioned to expose the internal structure and then cut vertically at the widest part. A scanning electron microscope (SEM300, tungsten filament, CIQTEK Co., Ltd., Hefei, China) was used to observe the cross-section, and a built-in ruler function was utilized to accurately measure leaf thickness. The sample was precisely excised into tissue blocks with dimensions of 0.5 cm × 0.5 cm × 0.5 cm. Subsequently, the samples were subjected to fixation using a 2.5% glutaraldehyde solution at either room temperature or 4 °C. For the dehydration process, a graded series of acetone solutions was employed, with concentrations sequentially increasing from 30%, 50%, 70%, 80%, 95% to 100%. Each dehydration step was maintained for a duration of 15–20 min. To achieve critical-point drying, the samples were first treated with isoamyl acetate to replace the 100% ethanol. This replacement was conducted at room temperature for a minimum of 20 min and repeated twice to ensure complete substitution. Following this, the samples were placed in a chamber containing supercritical fluid ($CO_2$). The residual moisture within the samples was then removed by the supercritical fluid during the critical-point drying procedure. The same equipment was used to measure stomatal density, length, width, and trichome density, stem diameter, and length. All data were obtained manually, with stomatal and trichome densities expressed as counts per square millimeter (no. $mm^{-2}$) and other parameters expressed in micrometers. Finally, the SEM300 was used to image the microstructure of the adaxial and abaxial surface of the same leaf within a 50 μm range, and the images were saved.

## Statistical analysis

Data were analyzed by SPSS 26.0 (IBM Corp., Armonk, NY, USA). The normality and homogeneity of variance of the data were tested *via* analysis of variance (ANOVA) to

evaluate the effects of different forage species on the quantity of epiphytic LAB and surface structure. An experimental model was constructed using the overall mean, treatment factors, and residuals (*Li et al., 2018*). Data with a *P*-value of <0.05 were considered significant. The systematic clustering analysis of different forage species was conducted on SPSS 26.0. Graphing was performed using the Platform Personalbio Genscloud and Origin 2024.

## RESULTS

### Effects of forage species on leaf morphological properties and epiphytic LAB counts

Significant differences were found among the different forage species in terms of the epiphytic LAB counts, leaf length, leaf width, leaf thickness, and cell wall thickness ($P < 0.001$) (Table 2). White goosefoot and beggarticks had the highest LAB counts (5.22 $\log_{10}$ CFU g$^{-1}$ FM and 4.83 $\log_{10}$ CFU g$^{-1}$ FM, respectively) but shortest leaf lengths (5.06 cm and 4.97 cm). Although maize and sweet potato had lower LAB counts (3.37 $\log_{10}$ CFU g$^{-1}$ FM and 3.38 $\log_{10}$ CFU g$^{-1}$ FM, respectively), their leaf widths were significantly greater than those of the other crops except for mallow ($P < 0.001$). The difference in leaf thickness among the 12 crops reached 781 $\mu$m, with chili pepper having the thickest leaves (812 $\mu$m) and beggarticks (31.0 $\mu$m) and perilla (50.6 $\mu$m) having the thinnest leaves. In terms of cell wall thickness, white goosefoot had significantly thinner cell walls, compared to all other crops ($P < 0.001$) except for common bean.

### Effects of forage species on the contact angle of leaves

The contact angles of the leaves from different forage species showed significant differences on both the adaxial and abaxial surfaces ($P < 0.001$) (Table 3). Beggarticks, common bean, and white goosefoot had significantly higher contact angles than the other crops ($P < 0.001$). On the adaxial surface, common bean had the highest contact angle (157°), whereas mallow had the lowest (58.2°). The contact angles of chili pepper (92.3°), tomato (83.4°), peanut (95.9°), and potato (77.3°) were within the medium range. On the abaxial surface, common bean had the highest contact angle (159°), whereas morning glory had the lowest (49.1°). Other crops, such as chili pepper (83.0°), tomato (91.0°), and potato (83.3°), also had relatively high contact angles. Overall, the trends in contact angles on both the adaxial and abaxial surfaces were similar for all crops.

### Effects of forage species on stomata and trichomes on the surfaces of leaves

The stomatal length, width, and density on the adaxial and abaxial surfaces of leaves in different forage species differed significantly ($P < 0.001$) (Table 4). On the adaxial and abaxial surfaces of leaves, the stomatal lengths of maize (48.4 $\mu$m and 50.6 $\mu$m, respectively) and potato (39.4 $\mu$m and 39.5 $\mu$m, respectively) were significantly greater than those of the other crops ($P < 0.001$). In contrast, the stomatal length on the adaxial and abaxial surfaces of morning glory (19.7 $\mu$m) and perilla (21.5 $\mu$m) was the shortest, respectively. In terms of stomatal width, common bean (21.7 $\mu$m) and potato (20.0 $\mu$m) had significantly wider

**Table 2  Measurements of leaf structure and epiphytic lactic acid bacteria of different forage species ($n = 36$).**

| Forage species | Lactic acid bacteria (log$_{10}$ CFU g$^{-1}$ FM) | Leaf length (cm) | Leaf width (cm) | Leaf thickness (μm) | Cell wall thickness (μm) |
|---|---|---|---|---|---|
| Maize | 3.37 ± 0.27d | 59.0 ± 3.09a | 8.63 ± 0.08b | 50.6 ± 2.10gh | 5.40 ± 0.41bc |
| Beggarticks | 4.83 ± 0.12ab | 4.97 ± 0.13f | 3.02 ± 0.16f | 87.4 ± 3.26fhg | 4.70 ± 0.36c |
| White goosefoot | 5.22 ± 0.02a | 5.06 ± 0.18f | 4.72 ± 0.23de | 110 ± 1.47efg | 2.09 ± 0.22f |
| Common bean | 4.35 ± 0.05d | 7.95 ± 0.42cd | 6.66 ± 0.44c | 413 ± 11.9c | 3.00 ± 0.21ef |
| Morning glory | 4.59 ± 0.07b | 7.85 ± 0.32cd | 6.24 ± 0.27c | 168 ± 9.54de | 6.55 ± 0.49a |
| Perilla | 3.85 ± 0.32cd | 8.44 ± 0.46c | 6.33 ± 0.39c | 31.0 ± 3.77 h | 5.90 ± 0.32ab |
| Tomato | 3.56 ± 0.24d | 8.53 ± 0.53c | 5.44 ± 0.35cd | 536 ± 38.4b | 4.45 ± 0.35cd |
| Chili pepper | 3.57 ± 0.24d | 6.97 ± 0.39cde | 3.30 ± 0.20f | 812 ± 62.5a | 5.30 ± 0.43bc |
| Sweet potato | 3.38 ± 0.27d | 11.1 ± 0.89b | 9.20 ± 0.97b | 187 ± 8.74d | 6.75 ± 0.25a |
| Peanut | 3.76 ± 0.31cd | 5.75 ± 0.26ef | 2.92 ± 0.14f | 129 ± 5.35def | 5.25 ± 0.34bc |
| Potato | 3.86 ± 0.11cd | 6.47 ± 0.27def | 3.83 ± 0.21ef | 51.8 ± 2.21gh | 3.70 ± 0.31de |
| Malva verticillata | 3.61 ± 0.12d | 11.5 ± 0.78b | 12.5 ± 0.81a | 110 ± 2.89efg | 3.25 ± 0.23e |
| SEM | 0.11 | 0.84 | 0.28 | 36.8 | 0.15 |
| P | <0.001 | <0.001 | <0.001 | <0.001 | <0.001 |

**Notes.**
Different lowercase letters in the same column represent significant difference among forage species ($P < 0.05$), with detailed information on which species differ significantly provided in the main text. SEM, standard error of the means, it measured the degree of difference between the sample mean and the population mean.

**Table 3  Measurements of leaf contact angle of different forage species ($n = 36$).**

| Forage species | Adaxial leaf surfaces (°) | Abaxial leaf surfaces (°) |
|---|---|---|
| Maize | 82.4 ± 2.43ef | 67.7 ± 1.25f |
| Beggarticks | 145 ± 2.73b | 136 ± 2.28b |
| White goosefoot | 104 ± 1.63c | 102 ± 1.28c |
| Common bean | 157 ± 2.01a | 159 ± 1.11a |
| Morning glory | 59.2 ± 1.58 h | 49.1 ± 1.64 h |
| Perilla | 72.1 ± 1.89 g | 71.4 ± 1.53f |
| Tomato | 83.4 ± 2.16e | 91.0 ± 1.29d |
| Chili pepper | 92.3 ± 2.11d | 83.0 ± 1.65e |
| Sweet potato | 80.4 ± 1.50ef | 79.3 ± 1.95e |
| Peanut | 95.9 ± 0.54d | 81.7 ± 1.50e |
| Potato | 77.3 ± 1.90f | 83.3 ± 1.08e |
| Malva verticillata | 58.2 ± 1.23 h | 59.8 ± 2.19 g |
| SEM | 2.75 | 2.77 |
| P | <0.001 | <0.001 |

**Notes.**
Different lowercase letters in the same column represent significant difference among forage species ($P < 0.05$), with detailed information on which species differ significantly provided in the main text. SEM, standard error of the means, it measured the degree of difference between the sample mean and the population mean.

**Table 4  Measurements of leaf stomatal of different forage species ($n = 36$).**

| Forage species | Stomatal on adaxial leaf surfaces | | | Stomatal on abaxial leaf surfaces | | |
|---|---|---|---|---|---|---|
| | Length (μm) | Width (μm) | Density (no. mm$^{-2}$) | Length (μm) | Width (μm) | Density (no. mm$^{-2}$) |
| Maize | 48.4 ± 0.80a | 11.7 ± 0.55e | 51.9 ± 2.45c | 50.6 ± 0.89a | 9.88 ± 0.28d | 63.1 ± 9.54d |
| Beggarticks | 28.2 ± 1.58ef | 17.0 ± 1.84b | 50.3 ± 3.54c | 32.6 ± 0.44c | 16.7 ± 0.62c | 68.6 ± 4.23cd |
| White goosefoot | 27.4 ± 0.41efg | 13.9 ± 0.61de | 17.3 ± 1.35e | 31.0 ± 0.80c | 16.5 ± 0.68c | 67.4 ± 3.97cd |
| Common bean | 36.8 ± 1.37c | 21.7 ± 0.99a | 66.3 ± 16.9b | 22.1 ± 0.88fg | 15.4 ± 1.28c | 150 ± 8.19b |
| Morning glory | 19.7 ± 0.30j | 13.6 ± 0.63de | 105 ± 3.59a | 24.3 ± 0.57e | 16.1 ± 0.67c | 179 ± 12.7a |
| Perilla | 21.2 ± 0.45ij | 14.5 ± 0.59cd | 33.6 ± 5.31d | 21.5 ± 0.30 g | 15.9 ± 0.24c | 179 ± 8.98a |
| Tomato | 32.7 ± 0.60d | 17.1 ± 0.38b | 27.7 ± 3.32de | 27.7 ± 0.37d | 16.6 ± 0.25c | 24.8 ± 4.91e |
| Chili pepper | 29.3 ± 0.67e | 16.7 ± 0.54bc | 25.1 ± 1.78de | 32.3 ± 0.35c | 21.9 ± 0.40b | 93.7 ± 1.80c |
| Sweet potato | 26.3 ± 0.57fg | 17.0 ± 0.38b | 29.8 ± 1.14de | 32.8 ± 0.47c | 21.0 ± 0.58b | 91.6 ± 5.83c |
| Peanut | 23.5 ± 0.56hi | 12.8 ± 0.56de | 103 ± 2.99a | 23.8 ± 0.47ef | 15.7 ± 0.73c | 73.9 ± 4.77cd |
| Potato | 39.4 ± 0.72b | 20.0 ± 0.98a | 15.2 ± 2.18e | 39.5 ± 1.03b | 23.9 ± 0.80a | 55.2 ± 3.68d |
| Malva verticillata | 25.4 ± 1.18gh | 16.8 ± 0.40bc | 50.3 ± 8.02c | 23.7 ± 0.68ef | 15.8 ± 0.73c | 82.5 ± 4.06cd |
| SEM | 0.76 | 0.34 | 3.12 | 0.77 | 0.37 | 5.76 |
| P | <0.001 | <0.001 | <0.001 | <0.001 | <0.001 | <0.001 |

**Notes.**

Different lowercase letters in the same column represent significant difference among forage species ($P < 0.05$), with detailed information on which species differ significantly provided in the main text. SEM, standard error of the means, it measured the degree of difference between the sample mean and the population mean.

stomata on the adaxial surfaces compared to the other crops ($P < 0.001$), while maize had the smallest stomata at 11.7 μm. On the abaxial surfaces, potato had the widest stomata (23.9 μm) while maize (9.88 μm) had the smallest stomata ($P < 0.001$). Moreover, morning glory (105 no. mm$^{-2}$) and peanut (103 no. mm$^{-2}$) had the highest stomatal density on the adaxial surfaces ($P < 0.001$), while morning glory (179 no. mm$^{-2}$) and perilla (179 no. mm$^{-2}$) had the highest stomatal density on the abaxial surfaces ($P < 0.001$).

On the adaxial surfaces of the leaves (Table 5), maize had significantly longer trichomes (1,589 μm) than the other crops, while chili pepper had significantly shorter trichomes (13.9 μm) and stem diameter (8.30 μm) ($P < 0.001$). Maize had the lowest trichome density (0.44 no. mm$^{-2}$). On the abaxial surfaces, perilla (29.6 μm), chili pepper (49.1 μm), and maize (59.1 μm) had shorter trichomes and exhibited similar trends in stem diameter. White goosefoot (0.85 no. mm$^{-2}$) and sweet potato (0.23 no. mm$^{-2}$) had lower trichome density on the abaxial surfaces. On both the adaxial and abaxial surfaces, tomatoes (34.4 no. mm$^{-2}$ and 61.3 no. mm$^{-2}$, respectively) had the highest trichome density among all crops ($P < 0.001$).

## Effects of leaf sides on the morphological structure of leaves

The contact angles on the adaxial (92.3°) and abaxial (88.6°) surfaces of leaves showed minimal differences ($P > 0.05$). Although the leaf surfaces had no significant effect on stomatal length ($P > 0.05$), stomatal width and density on the adaxial surfaces were significantly higher than those on the abaxial surfaces of the same leaves ($P < 0.05$). Regarding trichome properties, trichome length, stem diameter, and density on the adaxial

**Table 5  Measurements of leaf trichomes of different forage species ($n = 36$).**

| Forage species | Trichomes on adaxial leaf surfaces | | | Trichomes on abaxial leaf surfaces | | |
|---|---|---|---|---|---|---|
| | Length (μm) | Stem diameter (μm) | Density (no. mm$^{-2}$) | Length (μm) | Stem diameter (μm) | Density (no. mm$^{-2}$) |
| Maize | 1589 ± 114a | 45.5 ± 3.66d | 0.44 ± 0.05c | 59.1 ± 1.89ef | 9.41 ± 0.30d | 13.3 ± 1.01b |
| Beggarticks | 202 ± 10.0d | 27.7 ± 2.06e | 2.26 ± 0.43c | 188 ± 32.1cd | 49.2 ± 18.0b | 1.15 ± 0.27e |
| White goosefoot | 193 ± 71.3d | 9.18 ± 2.60f | 10.3 ± 2.49b | 180 ± 44.9cd | 19.8 ± 6.14cd | 0.85 ± 0.19e |
| Common bean | 323 ± 47.7c | 87.1 ± 5.93a | 0.77 ± 0.06c | 66.1 ± 4.12ef | 21.5 ± 2.34cd | 14.8 ± 0.99b |
| Morning glory | 364 ± 22.0c | 47.9 ± 1.43d | 1.29 ± 0.14c | 307 ± 37.2b | 49.1 ± 3.88b | 2.17 ± 0.13e |
| Perilla | 168 ± 17.9d | 60.9 ± 3.36cd | 2.17 ± 0.17c | 29.6 ± 2.98f | 19.3 ± 3.13cd | 3.23 ± 0.60de |
| Tomato | 100 ± 6.69de | 24.9 ± 1.36e | 34.4 ± 1.87a | 114 ± 8.71de | 22.9 ± 0.74cd | 61.3 ± 2.88a |
| Chili pepper | 13.9 ± 2.17e | 8.30 ± 1.01f | 14.0 ± 2.97b | 49.1 ± 3.25ef | 20.2 ± 1.24cd | 3.04 ± 0.35de |
| Sweet potato | 668 ± 59.0b | 70.0 ± 6.52bc | 0.57 ± 0.04c | 319 ± 93.2b | 25.3 ± 5.50cd | 0.23 ± 0.03e |
| Peanut | 43.2 ± 2.31e | 17.3 ± 0.87ef | 4.15 ± 0.23c | 161 ± 9.79cd | 17.8 ± 1.06cd | 5.99 ± 0.65cd |
| Potato | 344 ± 68.3c | 82.6 ± 10.4ab | 0.50 ± 0.06c | 197 ± 10.3c | 40.8 ± 2.17bc | 6.63 ± 0.33c |
| Malva verticillata | 372 ± 32.4c | 98.0 ± 6.89a | 0.71 ± 0.13c | 436 ± 37.0a | 89.3 ± 1.12a | 1.09 ± 0.09e |
| SEM | 31.9 | 3.14 | 0.98 | 13.7 | 2.83 | 1.56 |
| *P* | <0.001 | <0.001 | <0.001 | <0.001 | <0.001 | <0.001 |

**Notes.**
Different lowercase letters in the same column represent significant difference among forage species ($P < 0.05$), with detailed information on which species differ significantly provided in the main text. SEM, standard error of the means, i t measured the degree of difference between the sample mean and the population mean.

surfaces were 140 μm and 15.9 μm higher and 36.2% lower ($P < 0.05$), respectively, than those on the abaxial surfaces of the same leaves.

The microstructures of leaf surfaces (50 μm scale) showed significant differences among forage species, primarily in the morphology and arrangement of epidermal cells and properties of surface appendages. Maize exhibited a relatively smooth trait on the adaxial surfaces of leaves (Fig. 1A), whereas beggarticks displayed a rough surfaces covered with numerous small protrusions (Fig. 1B). Additionally, structural differences between the adaxial and abaxial surfaces of the same leaves were prominent. For instance, the adaxial surfaces of white goosefoot appeared smooth with nearly no visible trichomes (Fig. 1C), whereas the abaxial surfaces of this leaf had dense and longer trichomes, indicating well-developed structures (Fig. 2). For morning glory, the adaxial surfaces had a thicker wax layer and presented a smooth texture and higher reflectivity (Fig. 1E), whereas the abaxial surfaces had a thinner wax layer, increased roughness, and more complex structures (Fig. 2E). Perilla exhibited high roughness on both the adaxial and abaxial surfaces of the same leaves (Figs. 1F and 2F). The adaxial surfaces of the chili pepper were relatively smooth (Fig. 1H), whereas the abaxial surfaces were densely covered with glandular trichomes (Fig. 2H). In summary, different crops exhibited significant variations in stomatal density, epidermal appendage distribution, and epidermal roughness, with the most pronounced structural differences observed in white goosefoot and morning glory.

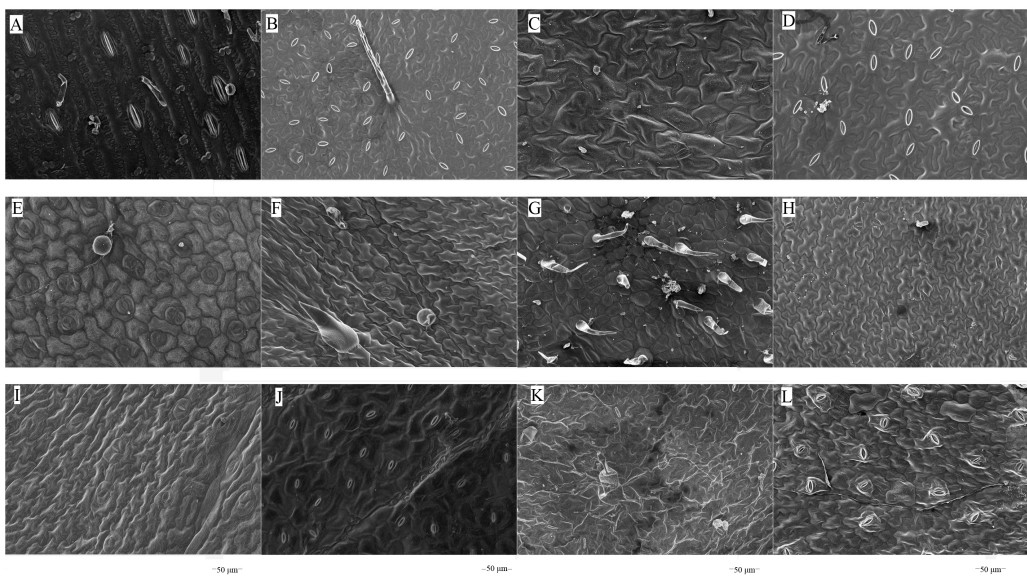

**Figure 1** **Scanning electron microscopy images on adaxial leaf surfaces of different forage species.**
Note: (A) Maize; (B) beggarticks; (C) white goosefoot; (D) common bean; (E) morning glory; (F) perilla;
(G) tomato; (G) chili pepper; (I) sweet potato; (J) peanut; (K) potato; (L) malva verticillata.

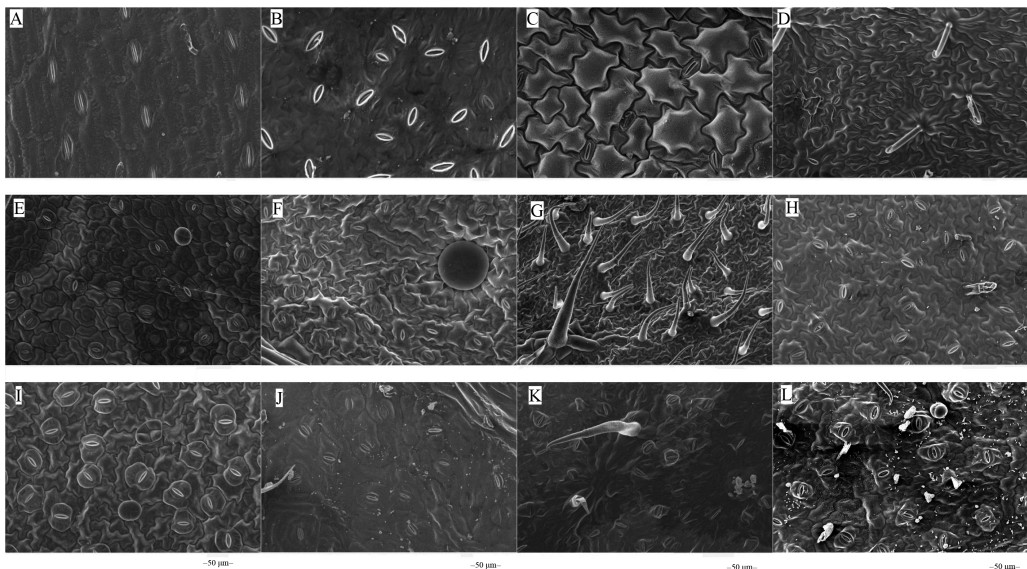

**Figure 2** **Scanning electron microscopy images on abaxial leaf surfaces of different forage species.**
Note: (A) Maize; (B) beggarticks; (C) white goosefoot; (D) common bean; (E) morning glory; (F) perilla;
(G) tomato; (H) chili pepper; (I) sweet potato; (J) peanut; (K) potato; (L) malva verticillata.

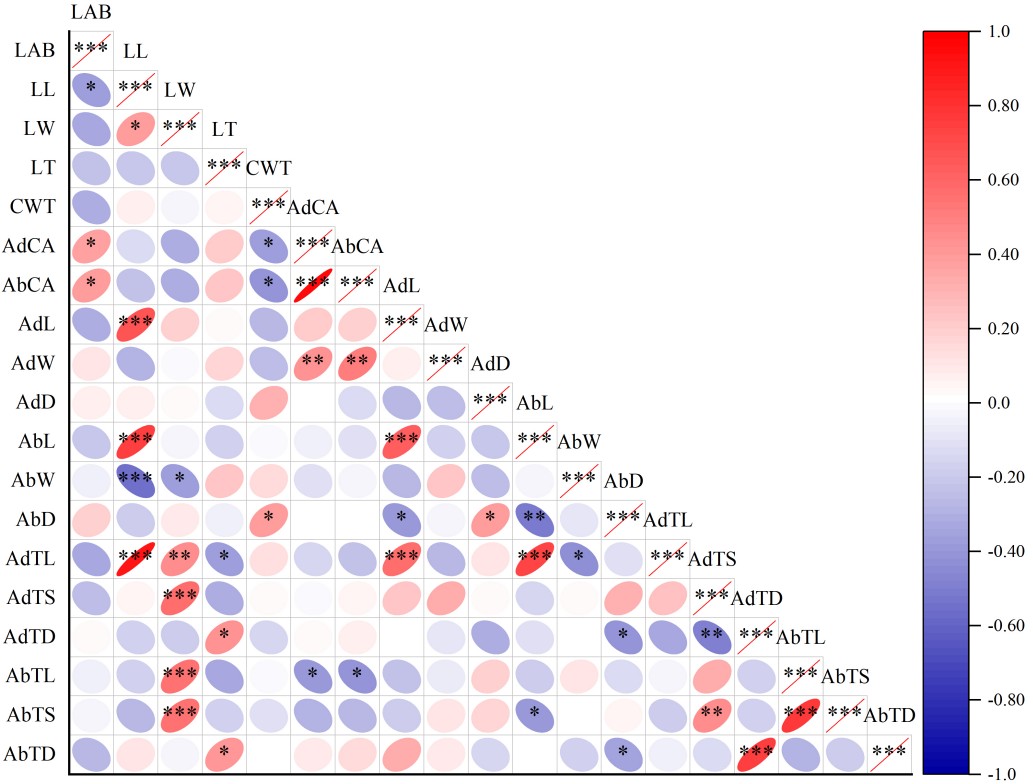

**Figure 3** **Correlation plot of Pearson among leaf surface structure and lactic acid bacteria counts.**
Note: Asterisks indicate significant differences at $P < 0.05$ (*), $P < 0.01$ (**), and $P < 0.001$ (***),
respectively. LAB, lactic acid bacteria; LL, leaf length; LW, leaf width; LT, leaf thickness; CWT, cell wall
thickness; AdCA, contact angle on adaxial leaf surfaces; AbCA, contact angle on abaxial leaf surfaces; AdL,
stomatal length on adaxial leaf surfaces; AdW, stomatal width on adaxial leaf surfaces; AdD, stomatal den-
sity on adaxial leaf surfaces; AbL, stomatal length on abaxial leaf surfaces; AbW, stomatal width on abaxial
leaf surfaces; AbD, stomatal density on abaxial leaf surfaces; AdTL, trichomes length on adaxial leaf sur-
faces; AdTS, trichomes stem diameter on adaxial leaf surfaces; AdTD, trichomes density on adaxial leaf
surfaces; AbTL, trichomes length on abaxial leaf surfaces; AbTS, trichomes stem diameter on abaxial leaf
surfaces; AbTD, trichomes density on abaxial leaf surfaces.

## Correlation and clustering analyses of leaf morphological structures and LAB in forage crops

The LAB counts was significantly positively correlated with the contact angle of the adaxial
and abaxial surfaces of leaves ($P < 0.05$) and significantly negatively correlated with the leaf
length ($P < 0.05$). No significant correlations were observed between LAB counts and other
structural factors ($P > 0.05$) (Fig. 3). Further linear regression analysis revealed that the
coefficients of determination ($R^2$) between LAB counts and contact angles on the adaxial
and abaxial surfaces of leaves were 0.1424 (Fig. 4A) and 0.175 (Fig. 4D), respectively,
whereas the $R^2$ values between stomatal and trichome densities on both surfaces were all
below 0.07, indicating a weaker explanatory power for these factors (Fig. 4). Cluster analysis
based on the LAB counts and leaf morphological structural properties grouped morning

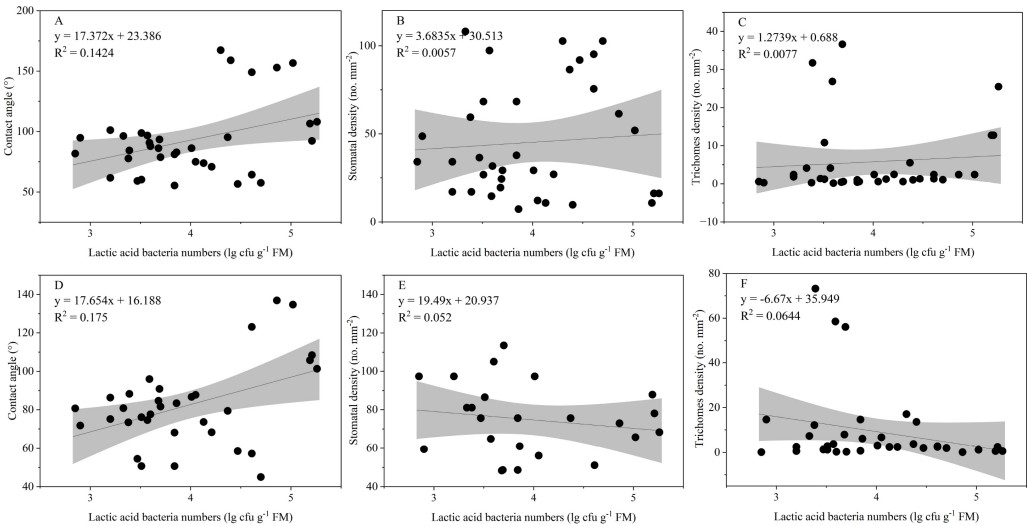

**Figure 4 Influence of contact angle, stomatal, and trichomes densities on the number of epiphytic lactic acid bacteria.** Note: Projection represents a 95% confidence interval. (A), (B), and (C) Represents the contact angle, stomatal, and densities on adaxial leaf surfaces of the leaf, respectively; (D), (E), and (F) represents the contact angle, stomatal, and densities on abaxial leaf surfaces of the leaf, respectively.

glory and common bean into a single cluster, tomato, and chili pepper into another cluster, and the remaining eight crops into a third cluster (Fig. 5).

## DISCUSSION

### Effects of leaf morphological features on the epiphytic LAB counts

Morphological features of plant leaves, such as the leaf vein arrangement and leaf shape, directly influence the LAB distribution. Different vein patterns (*e.g.*, parallel, pinnate, and palmate) and leaf shapes (*e.g.*, needle, heart, and oval shapes) alter the microenvironment on the leaf surfaces (*Doan et al., 2020*), thus affecting the attachment and growth of LAB. Pinnate and palmate veins typically promote LAB attachment by providing a larger leaf area and a more complex venation structure (*Sack & Scoffoni, 2013*). Additionally, leaves with pinnate and palmate veins present higher surface roughness and thus are more effective at capturing airborne moisture, which further promotes LAB growth and reproduction. In addition, the relatively high rainfall during the sampling time (August) further provided moisture conditions conducive to the proliferation of LAB in this study. However, parallel veins may lead to rapid moisture loss from the leaf surfaces, increasing the risk of inadequate water availability for LAB colonization. Previous studies have confirmed that complex leaf epidermal structures promote the attachment and growth of LAB on leaf surfaces (*Tang et al., 2023*). Moreover, needle-shaped leaves, which have a smaller surface area and lower roughness of leaves, may be less conducive to bacterial attachment, whereas heart- and oval-shaped leaves, which present a larger leaf surface area and more complex morphological structure on the surfaces of leaves, can more effectively adsorb LAB.

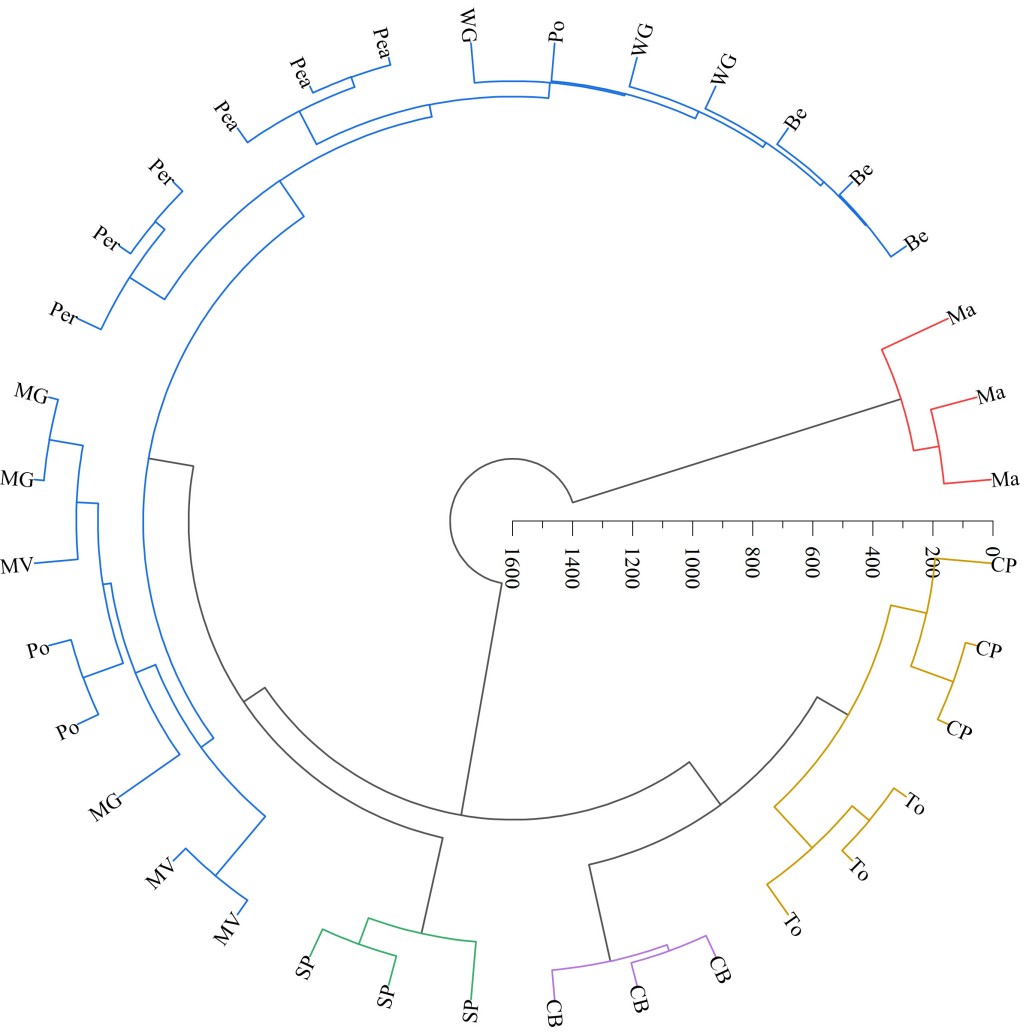

**Figure 5  Systematic clustering analysis of different forage species.** Note: The same color indicates that forage species were classified into one category. Ma, maize; Be, beggarticks; WG, white goosefoot; CB, common bean; MG, morning glory; Per, perilla; To, tomato; CP, chili pepper; SP, sweet potato; Pea, peanut; Po, potato; MV, malva verticillata.

In this study, both beggarticks and white goosefoot had pinnate venation, with white goosefoot presenting lanceolate leaves and beggarticks presenting pinnate leaves, and both showed a higher LAB counts on the surfaces of the leaves. The higher LAB counts on the surfaces of leaves in white goosefoot may have been related to their thinner leaves. Typically, thinner leaves can better absorb external water and nutrients, providing ample growth resources for LAB (*Boanares et al., 2018*). Furthermore, thinner leaves have a higher surface-area-to-volume ratio, which facilitates gas exchange, particularly that of oxygen and carbon dioxide (*Harrison et al., 2019*), thus providing an ideal environment for the growth of aerobic LAB (*Tang et al., 2023*). Unfortunately, this study did not analyze the nutrient and enzyme activity on the surfaces or inside of the leaves in these forage species.

## Effects of contact angle on the LAB counts

Contact angle is a critical physical parameter for evaluating the hydrophilic or hydrophobic nature of forage surfaces (*Miller et al., 2019*). This parameter significantly impacts the initial adhesion, proliferation, and metabolic activity of microorganisms. A smaller contact angle (<90°) indicates stronger hydrophilicity of the surfaces, which favors the adhesion of hydrophilic microorganisms. In contrast, a larger contact angle (>90°) suggests a more hydrophobic environment suitable for the adhesion and growth of hydrophobic microorganisms. This selective adhesion plays a pivotal role in shaping the structure and functional dynamics of the microbial communities. In this study, the leaves of beggarticks, white goosefoot, and common beans exhibited larger contact angles (Table 3), supporting a higher LAB counts (Table 2). LAB on leaf surfaces generally exhibits high carbon-to-nitrogen ratios, leading to pronounced hydrophobicity (*Chasoy, Chairez & Durán-Páramo, 2020*). Furthermore, LAB often faces environmental stresses such as low nutrient availability, limited water resources, UV radiation, oxidative stress, and dramatic temperature fluctuations under natural conditions (*Lindow & Brandl, 2003*). To adapt to these extreme conditions, LAB enhances extracellular polysaccharide synthesis and forms protective biofilms that increase their environmental adaptability and survival rates (*Nwodo, Green & Okoh, 2012*). Consequently, LAB populations on leaf surfaces can remain high, even with larger contact angles.

However, when the contact angle was small (<30°), the adhesion efficiency of LAB could increase by 40% to 50% (*Xiong et al., 2018*). This effect was crucial for optimizing the silage fermentation process (*Tan et al., 2022*) because smaller contact angles promote LAB colonization on leaf surfaces and significantly suppress competition from harmful microorganisms, such as yeast and *Clostridium* (*Lin et al., 2022*). Additionally, previous studies have demonstrated that leaf surfaces exhibit typical superhydrophobicity when the contact angle exceeds 150°, owing to the synergistic effects of micro-scale papillary structures and wax layers. Such surfaces cause water droplets to roll off in bead-like forms, thereby achieving self-cleaning effects that effectively remove dust and bacterial particles (*Shi, Wang & Li, 2011*). This superhydrophobicity further restricts bacterial adhesion to the leaf surfaces.

In summary, the contact angle of the leaf surfaces regulates water film formation and distribution, directly influencing LAB adhesion, dispersal behavior, and community structure. Variations in the contact angle determine the preferential attachment of specific microorganisms, thereby significantly affecting the dynamic balance of LAB communities on plant leaves and the spread of plant diseases. This finding provides a theoretical basis for optimizing the microecological environment of plant leaves.

## Effects of stomatal and trichome characteristics on the LAB counts

The population of LAB attached to forage is generally below 5.0 $\log_{10}$ CFU $g^{-1}$ FM (*Cai et al., 2020*; *Chen, Dong & Zhang, 2021*), while the population on the leaf surfaces typically does not exceed 4.20 $\log_{10}$ CFU $g^{-1}$ FM (*Wu et al., 2023*). These findings are consistent with the results of the present study, indicating that LAB on forage surfaces primarily colonize the leaf surfaces. Furthermore, the stomata and trichomes on plant

surfaces significantly influence the attachment, growth, and community structure of LAB by modulating the microenvironment, including moisture content, nutrient availability, and gas exchange. Specifically, larger stomata provide convenient entry pathways for bacterial colonization (*Koch & Barthlott, 2009*). Previous studies have shown that when the stomatal length exceeds 15 μm, the bacterial infection rate can increase by 20% to 30% (*Chen et al., 2023*). In this study, the stomatal length of all forage samples exceeded 15 μm, providing ideal conditions for LAB attachment and facilitating their entry into plant tissues. Additionally, higher stomatal density offers more attachment sites and promotes a micro-moist environment, further promoting the colonization and proliferation of LAB. Under high humidity conditions, densely distributed stomata more readily form stable water films, which are crucial for LAB growth (*Guan et al., 2018*). These findings highlighted the critical regulatory roles of stomatal structure and distribution in shaping the dynamic growth patterns of LAB.

Plant trichomes are critical as physical barriers and microenvironment regulators in plant-microbe interactions. Typically, trichomes enhance the roughness of leaf surfaces, facilitating the initial attachment and colonization of microorganisms. When trichome lengths range between 50–100 μm, the attachment of *Bacillus* species increases significantly by 25%–30%; when trichome lengths are below 50 μm, the attachment efficiency decreases (*Zhang et al., 2022*). Although the trichome length of the forage species did not significantly influence the abundance of LAB in this study (Fig. 3), its contribution to the surface roughness of forage leaves was undeniable (Figs. 1 and 2). Previous studies have shown that trichomes may positively affect the abundance of other bacteria and yeast species (*Tang et al., 2021*), which may be related to the plant species and the specificity of LAB strains. The surface structures of different forage species exhibit significant variation and play a selective role in microbial attachment and community composition. LAB isolated from forage surfaces primarily include genera such as *Enterococcus*, *Lactococcus*, *Lactobacillus*, *Pediococcus*, *Leuconostoc*, and *Weissella* (*Yu, Leveau & Marco, 2020*). These LAB strains generally display strong aerotolerance, a trait primarily mediated through mechanisms such as the production of cytochrome d oxidase or non-enzymatic dismutation of hydrogen peroxide *via* $Mn^{2+}$ ions, thus mitigating oxidative stress (*Papadimitriou et al., 2016*). This characteristic explains the high abundance and structural diversity of LAB communities on forage leaf surfaces. LAB species exhibit varying nutritional requirements, while available nutrients on the leaf surface are relatively limited (*Chen, Dong & Zhang, 2021*). Furthermore, trichome density and structure significantly influence the degree of surface moisture retention. High-density trichomes (>500 hairs no. $mm^{-2}$) can extend surface moisture retention times by 2 to 3 fold, creating favorable conditions for water-dependent microorganisms, such as *Pseudomonas* (*Koch & Ensikat, 2008*). Simultaneously, trichomes contribute to the formation of micro-humid environments on leaf surfaces, thereby promoting the proliferation and metabolic activity of LAB (*Yuan et al., 2017*).

The results of this study showed that the adaxial surfaces of the leaves exhibited smoother characteristics and a relatively thicker wax layer (Fig. 1), whereas the abaxial surfaces of the leaves displayed a greater stomatal distribution and more epidermal hairs (Table 6). These morphological differences may lead to a reduced number of LAB on the adaxial surfaces

**Table 6 Measurements of leaf contact angle, stomatal, and trichomes of different area ($n = 72$).**

| Area | Contact angle (°) | Stomatal | | | Trichomes | | |
|---|---|---|---|---|---|---|---|
| | | Length (μm) | Width (μm) | Density (no. mm$^{-2}$) | Length (μm) | Stem diameter μm) | Density (no. mm$^{-2}$) |
| Adaxial | 92.3 ± 2.75 | 29.9 ± 0.76 | 16.1 ± 0.34b | 47.1 ± 3.12b | 308 ± 31.9a | 49.4 ± 3.14a | 6.17 ± 0.98b |
| Abaxial | 88.6 ± 2.77 | 30.2 ± 0.77 | 17.1 ± 0.37a | 101 ± 5.76a | 168 ± 13.7b | 33.5 ± 2.83b | 9.67 ± 1.56a |
| SEM | 1.95 | 0.54 | 0.25 | 3.63 | 18 | 2.17 | 0.93 |
| $P$ | 0.337 | 0.775 | 0.039 | <0.001 | <0.001 | <0.001 | 0.049 |

Notes.
Different lowercase letters in the same column represent significant difference between area ($P < 0.05$), with detailed information on which area differ significantly provided in the main text. SEM, standard error of the means, it measured the degree of difference between the sample mean and the population mean.

compared to the abaxial surfaces. Previous studies have also confirmed that the abaxial surfaces of the leaves provides more favorable environmental conditions for the survival of LAB owing to its rich nutrient reserves and relatively rough structure on the surfaces of leaves (*Tang et al., 2023*).

In conclusion, plant trichomes regulate microbial attachment, survival, and colonization by increasing surface roughness, creating humid microenvironments, and blocking ultraviolet radiation. Additionally, leaf morphological structures on the surfaces of leaves modulate water availability, gas exchange, and nutrient accessibility, thus playing a pivotal role in shaping the composition and dynamic balance of microbial communities on plant surfaces. These attributes provide a theoretical foundation for understanding plant-microbe interaction mechanisms and optimizing forage silage fermentation processes and improve the fermentation quality of silage. However, given the limitations of the linear regression model in this study, future research may need to employ data analysis methods such as principal component analysis or multivariate regression to further explore the correlations between the quantity and structure of LAB. In addition, this study has not conducted an in-depth analysis of the species and genera of lactic acid bacteria, which, to a certain extent, has indirectly affected our understanding of their fermentation potential in silage.

## CONCLUSION

Leaf morphological properties play important roles in regulating LAB abundance. Leaves with pinnate and palmate venation presented more complex vein structures and larger leaf areas, thus providing more micro-scale attachment points, promoting LAB colonization. Complex and rough leaf morphological structures effectively capture airborne moisture, forming a micro-humid environment that further optimizes the regulation of water flow, gas exchange, and ultraviolet radiation. Collectively, these factors create a favorable environment for the attachment, colonization, and proliferation of LAB. Therefore, plant leaves, such as those of white goosefoot, beggarticks, and morning glory, have a higher LAB population.

### Funding

This research was funded by Xingzhao Talent Support Program (2023) and Liupanshui Municipal Key Laboratory for Development and Utilization of Feed Resources (52020-2024-PT-02), Special Basic Cooperative Research Programs of Yunnan Provincial Undergraduate Universities' Association (202301BA070001-120), and High-Level Talents Introduction Project of Liupanshui Normal University (LPSSYKYJJ202306). The funders had no role in study design, data collection and analysis, decision to publish, or preparation of the manuscript.

### Grant Disclosures

The following grant information was disclosed by the authors:
Xingzhao Talent Support Program (2023).
Liupanshui Municipal Key Laboratory for Development and Utilization of Feed Resources: 52020-2024-PT-02.
Special Basic Cooperative Research Programs of Yunnan Provincial Undergraduate Universities' Association: (202301BA070001-120.
High-Level Talents Introduction Project of Liupanshui Normal University: LPSSYKYJJ202306.

### Competing Interests

The authors declare there are no competing interests.

### Author Contributions

- Dan Wu conceived and designed the experiments, analyzed the data, prepared figures and/or tables, authored or reviewed drafts of the article, and approved the final draft.
- Guicong Tang conceived and designed the experiments, analyzed the data, prepared figures and/or tables, authored or reviewed drafts of the article, and approved the final draft.
- Gaofeng Liu performed the experiments, authored or reviewed drafts of the article, and approved the final draft.
- Ting Sun performed the experiments, authored or reviewed drafts of the article, and approved the final draft.
- Jinmei Yang performed the experiments, authored or reviewed drafts of the article, and approved the final draft.
- Guojian Tang conceived and designed the experiments, analyzed the data, prepared figures and/or tables, authored or reviewed drafts of the article, and approved the final draft.
- Liuxing Xu conceived and designed the experiments, analyzed the data, prepared figures and/or tables, authored or reviewed drafts of the article, and approved the final draft.

### Data Availability

The raw measurements are available in the Supplementary Files.

## Supplemental Information

Supplemental information for this article can be found online at http://dx.doi.org/10.7717/peerj.20028#supplemental-information.

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
