# Peer review of "Influence of leaf morphological properties on epiphytic lactic acid bacteria counts in forage crops"

_PeerJ, doi:10.7717/peerj.20028_

## Round 0.1 · original submission · Major Revisions

Please address all reviewers' comments.

Reviewer 1 ·

Basic reporting

The authors identify an interesting topic, the attachment characteristics of lactic acid bacteria (LAB) to forage leaves. However, the manuscript should be improved to enhance clarity. While background is provided, the flow should be improved. The scale bar in the figures is not clear. The table caption should be clearer.
1. Units in Abstract: The meaning of units such as "lg cfu g-1 FM" and "no. mm-2" in the Abstract should be explicitly defined to ensure immediate comprehension by a broad readership.
2. Background and Impact: The Abstract lacks a clear articulation of the broader background and potential impact of this research, which should be emphasized to contextualize the study's significance.
3. Abstract Content: The Abstract appears to contain excessive detail regarding the specific results. Consider moving some of these findings to the Introduction to provide a more focused overview in the Abstract.
4. Introduction Logic: The logical flow within the first paragraph of the Introduction is unclear and requires improved writing for better coherence and understanding of the research context.
5. Incomplete Sentence (Line 69): The sentence "whereas a thicker epidermis …" on Line 69 is incomplete and confusing. Please review and complete this sentence to convey the intended meaning.
6. Attachment vs. Count: The Introduction primarily discusses LAB attachment to forage leaves, but later shifts focus to LAB counts. The relationship between these two parameters (attachment and count) needs to be explicitly stated and justified.
7. Missing Information in Introduction: The Introduction should briefly mention the main methods employed and the key findings of the study to provide a more comprehensive overview for the reader.
8. Presentation Error (Line 127): The heading "2.5 Statistical analysis" on Line 127 appears to be a formatting or presentation error. Please correct this.
9. Confusing Phrase (Line 135): The phrase "Influence of forage species on leaf length, leaf width, leaf thickness, cell wall thickness, and epiphytic LAB counts" on Line 135 is conceptually confusing. Rephrase this to clearly indicate how forage species influence these structural parameters and consequently impact LAB counts.
10. Table Details:
The sample numbers (n) for each experimental group in Tables 2-5 should be clearly stated in the respective table captions.
A schematic diagram illustrating the measurement methods employed would enhance the clarity of the experimental procedure.
While the caption mentions that different lowercase letters indicate significant differences, the meaning of these specific letters (e.g., which species are significantly different from others) is not provided. This information is essential for interpreting the tables.
The justification for presenting the Standard Error of the Mean (SEM) for different species and an explanation of the information this SEM provides in the context of the data should be included.
11. Unclear Meaning of "Location" (Line 178): The meaning of "location" in the context of "Influence of location on the leaf surface structure" (Line 178) is unclear and needs to be defined.

Experimental design

The research question is well defined and will fill a knowledge gap. But more details should be provided in the Methods.

1. Experiment Site Data: The provided data on the experiment site's average annual temperature (12.3°C) and annual total rainfall (682 mm) should be included in the Methods section. Justification for their potential impact on LAB attachment, colonization, and proliferation should also be discussed, potentially with relevant literature.
2. Leaf Structural Parameters (Line 113): The rationale for using leaf length and width instead of surface area as primary leaf structural parameters for LAB attachment should be clearly justified. Surface area seems more directly relevant to microbial colonization.
3. Dehydration Method (Line 121): The description of the leaf dehydration process (Line 121) lacks crucial details. Specify the dehydration conditions and address whether potential deformation and shrinkage of leaf structures during dehydration were considered and controlled for.
4. Representative Images: Including representative images of the forage leaf samples would significantly complement Table 1 and provide valuable visual context for the described structural parameters.

Validity of the findings

1. Surface Roughness Discussion (Line 218): The discussion on higher surface roughness (Line 218) is not supported by direct data on surface roughness, which is only qualitatively linked to trichomes later. Reorganize this section to either include quantitative data on surface roughness or focus the discussion solely on the observed trichome effects.
2. Unsupported Claim (Cell Wall Impact): The explanation regarding the cell wall's impact on LAB activity lacks experimental support within this study, as acknowledged by the authors. This unsubstantiated claim should be removed to maintain the scientific rigor of the manuscript.
3. Aerobic LAB Location (Line 231): The explanation linking thinner leaves, gas exchange, and an "ideal environment for the growth of aerobic LAB" (Line 231) raises questions about the location of LAB. If LAB are primarily located on the leaf surface (as suggested later), this explanation needs further clarification and literature support or experimental evidence to demonstrate LAB presence and activity inside the leaves.
4. Linear Regression Details (Line 326-327): More details about the final linear regression equation should be provided, including a figure illustrating the model's fit to the experimental dataset.
5. SEM images of cell walls should be provided.

Reviewer 2 ·

Basic reporting

Mostly clear, with some small areas that could be improved.
Literature references are sufficient, and the work ties in well to the field.
The structure and figure presentation are fine.
The attached file does not seem to contain the raw data, but perhaps I have misunderstood how to properly download.
Self-contained – seems a bit small of a project and limited result presentation.

Experimental design

Aims and scope seem to fit well.
The research question appears to be presented in the second paragraph in the introduction, but to me, it is unclear from the title and the abstract what is being investigated. Currently, I have the following as the research question: “Further in-depth research is needed to reveal the key factors influencing the attachment of LAB to leaves.”
The technical standard is sufficient.
The methods are described well enough to duplicate

Validity of the findings

Appears to be a new study looking at connections previously uninvestigated.
I have not been able to successfully acquire the underlying data.
In the conclusion, the authors indicate that leaf thickness and cell wall thickness play important roles in regulating LAB abundance, but a link was not shown in the correlation plot. The conclusions appear to better support the theory that was cited within the paper rather than the data presented or the original question.

Additional comments

My main concerns lie around the findings that relate specifically to your data. You sufficiently describe how the data was collected, but the analysis could be greatly strengthened. As currently written, it appears to be a well-collected experiment without much depth in the interpretation of the results. More could be explored or discussed about the correlation plot. Additionally, some interesting relationships appear in the correlation plot and could possibly be described or explored.

The title of the paper is also somewhat misleading, as it would suggest the surface LAB is being correlated with the surface structure. However, you do not consider the LAB on a single surface as compared to the structure of that surface. The title should be revised to better reflect the methods of the paper.

Specific comments are provided below.
o The abstract lists many data findings, but it would be improved if some additional interpretations of the data were included. This is shown in the second-to-last sentence, but additional information about the connection between leaf structure and LAB would be beneficial. I don’t think the main purpose of the paper is to connect similarities between the abaxial and adaxial sides of the leaf, but much time is spent on this in the abstract.
o The conclusions should support your data and findings. Currently, it seems like they summarize some previous work, but do not mention the main correlations that you found between your collected data and LAB counts.
o Check consistency in capitalization (Malva vs malva)
o Many sentences use unnecessary initial words. For example, “However” is used to introduce two sentences in a row on lines 41 and 43. The words “Notable” and “Thus” to introduce sentences later in this paragraph are also unnecessary.
o “higher surface roughness” should be rephrased, possibly to “greater surface roughness”
o In line 62, the phrase “more stable” is used to describe the attachment sites. Is this the accurate word, or are there just more sites in general for attachment? If the former, what makes it more stable?
o Line 68 – why the change between “leaves” and “forage” in “high-density stomatal leaves than on low-density stomatal forage”?
o Line 69 – What is meant by “whereas a thicker epidermis worldwide attachment”?
o Line 70 – What is leaf hardness? Is this the cell wall thickness?
o Is there a common name you can use for Malva verticillate instead of listing the Latin name twice?
o Line 101 – consider rewording “individuals of the same plant species” to “plants of the same species”
o Line 101 – does “with each crop sampled 10 times” indicate each species was sampled 10 times, each plant was sampled 10 times, each plot was sampled 10 times? Consider rewording for clarity.
o Line 111 – I would put “days” instead of “d” for units
o It would be interesting if you could compare the surface loading of LAB instead of the loading in the entire leaf – abaxial, adaxial, and interior. This could provide more specific information and could more successfully tie your observations together. I understand this is beyond the application of the current dataset.
o Is the cell wall thickness that was measured the epidermal cell wall thickness or the interior?
o Line 127 – “2.5 Statistical analysis” is this a heading?
o I’m unclear why LAB counts would be correlated with leaf length. Could some insight be provided as to why this might be the case?
o You refer to the contact angle, but did you also consider the leaf surface angle or the roughness of the epidermal cells?
o Line 158 – “leaf stomatal” should possibly be “leaf stomata”
o Line 162 – rephrase the sentence beginning with “In contrast” to fully describe one side before the other. Too many items to try to group together with “respectively”.
o Line 166 – 11.7 um on the adaxial side as well? This was unspecified.
o Line 183 – Are these values the averages, or do they refer to specific plants?
o Line 190 – “longer trichomes” longer than what?
o Line 191 – “thicker wax layer”, how was the wax measured? I may have missed this in the methods.

Reviewer 3 ·

Basic reporting

The manuscript addresses a timely and relevant topic by exploring how leaf surface morphology influences lactic acid bacteria (LAB) colonization, with implications for silage and livestock nutrition.

The manuscript is well referenced, though there is an over-reliance on citations from Tang et al., which reduces the breadth of scholarly context. More diverse and recent literature should be integrated to strengthen the scientific framing.

Figures and tables are comprehensive but need improvements for clarity:

Figure 3: Text is too small; label spacing and data grouping could be improved for readability.

Figures 1–2: SEM images are visually helpful but lack quantitative surface roughness metrics.

Table formatting could be improved by grouping species (e.g., cultivated vs. wild) and including visual indicators (e.g., shading or color codes) for significant findings.

Grammar and phrasing require careful revision. Examples:

Line 69: “whereas a thicker epidermis worldwide attachment” → should be “would hinder attachment.”

Use of “lg” instead of “log” for CFU expression throughout the abstract and tables.

Writing occasionally overstates findings (e.g., “strong capacity for LAB colonization”) without appropriately tempering based on statistical strength.

Suggestions:

Recommend professional English language editing to address recurring grammatical inconsistencies and improve scientific tone.

Revise captions to better explain the significance of figures, not just their contents.

Use consistent scientific terminology for measurements (e.g., "log₁₀ CFU g⁻¹", "μm").

Experimental design

The research question is relevant, and the comparative approach across multiple forage species is sound. However, methodological inconsistencies limit interpretability:

Plant maturity stages were not standardized across species. This introduces variation in physiology and surface traits that could confound LAB counts.

The sampling design lacks sufficient detail regarding randomization and independence of replicates.

LAB quantification methods are appropriate but missing critical control elements:

No negative/blank controls mentioned.

Anaerobic culture conditions are not clearly defined (e.g., gas composition, anaerobic chamber vs. jar).

Table 1 includes vague growth stage terms (e.g., “maturation stage”); recommend using standardized phenological descriptors (e.g., BBCH scale).

Suggestions:

Specify replication level and clarify how leaf samples were randomized across plants or plots.

Include details on how anaerobic conditions were maintained.

Align species sampling on similar growth stages or provide a rationale if maturity differs.

Validity of the findings

The data presented is comprehensive and carefully measured. However, key statistical results have limited explanatory power:

The R² values for contact angle vs. LAB counts (0.1424 and 0.175) indicate very weak linear correlations.

Additional variables such as stomatal and trichome characteristics showed even weaker associations (R² < 0.07), yet the discussion occasionally implies stronger effects.

Regression equations are presented, but multivariate relationships are not explored deeply (e.g., PCA or clustering could clarify trait groupings).

Interpretation of correlation is often overextended, with speculative claims not fully supported by data. For example:

Conflicting statements about whether high or low contact angles favor LAB adhesion (Lines 240–264).

Positive claims were made for traits (e.g., trichomes) that had no significant correlation with LAB abundance.

Limitations are acknowledged late in the discussion. These should be more clearly integrated throughout the narrative.

Suggestions:

Clearly state the limited predictive power of the regression models.

Include alternative statistical approaches (e.g., PCA, multivariate regression) or at least acknowledge their potential utility.

Add disclaimers where interpretations exceed what is statistically supported.

Move the statistical model summary (Lines 324–341) earlier in the discussion to frame findings appropriately.

Additional comments

Strengths:

The manuscript contributes valuable data on leaf microstructures (contact angle, trichome/stomatal features) and their potential role in microbial colonization.

The range of forage species tested is commendable and relevant to silage practice.

The authors make a commendable effort to link anatomical traits to microbial ecology.

Areas for Improvement:

Improve the logical flow and clarity of the discussion, particularly when interpreting contradictory findings (e.g., contact angle, trichomes).

Add contextual discussion on practical implications for silage fermentation, as silage is central to the motivation but not directly tested.

Consider elaborating how findings could guide forage crop selection or breeding for microbial compatibility.

Clarify whether LAB were identified beyond morphological characteristics; strain-level identification would greatly strengthen the conclusions.

---

## Round 0.2 · Minor Revisions

Please address all remaining minor comments.

Reviewer 1 ·

Basic reporting

All questions addressed.

Experimental design

All questions addressed.

Validity of the findings

All questions addressed.

Reviewer 2 ·

Basic reporting

Improvements have been made, but there are still some issues that can be addressed. Of the following, the first four were indicated in the first review, but there were no changes or explanations for decisions to not change.
- Check consistency in capitalization (Malva vs malva)
- Is there a common name you can use for malva verticillate instead of listing the Latin name twice?
- Does “with each crop sampled 10 times” indicate each species was sampled 10 times, each plant was sampled 10 times, each plot was sampled 10 times? Consider rewording for clarity.
- The phrase “more stable” is used to describe the attachment sites. Is this the accurate word or are there just more sites in general for attachment? If the former, what makes it more stable?
- “However” is still used to introduce two sentences in a row. The other introductory words appear to be improved.
- In the change to “larger leaf areas” instead of “larger leaf surface areas” I think the original made more sense. Is it not the surface structure having the effect rather than the overall size of the
- “whereas a thicker epidermis worldwide attachment” was changed to replace “worldwide” with “barely”. This sentence still does not make sense.

Experimental design

- Some additional context has been given to address the limitations of the linear regression and some suggestions for further analysis (PCA, multivariate regression) have been suggested, but not employed. The analysis of the data still feels somewhat superficial.

Validity of the findings

As was noted in the first review, the conclusions should support your data and findings. Currently, it seems like they summarize some previous work, but do not mention the main correlations that you found between your collected data and LAB counts.

Additional comments

The abstract still lists many individual data findings, and it may be more appropriate to reduce these in favor of a summary of the major findings. The sentence added at the end is an improvement.

Reviewer 3 ·

Basic reporting

Clarity: The revised version is clearly written with improved grammar and flow.

Structure: The manuscript follows a standard structure (Abstract, Introduction, Methods, Results, Discussion, Conclusion) and is well-organized.

Figures/Tables: Tables are informative, clearly labeled, and directly support the data interpretation. Figures were not included in the shared files, but the text references are appropriate.

Experimental design

Scientific Rigor: The study uses standard and reliable methods (SEM, contact angle measurement, LAB culturing), and replicates are clearly defined.

Reproducibility: Methods are described in detail; the revised version strengthens clarity on sampling and analysis protocols.

Control of Confounding Variables: While not all potential variables are discussed (e.g., leaf nutrient content or weather conditions at sampling), the authors acknowledge these limitations in the discussion.

Validity of the findings

Statistical Analysis: Appropriate ANOVA and regression were used. Authors candidly acknowledge the limited explanatory power of some findings (e.g., low R² values).

Support of Conclusions: The conclusions are well-aligned with data and limitations are discussed.

Novelty & Relevance: The study adds valuable understanding to how leaf morphology influences epiphytic LAB—important for silage and animal feed applications.

Additional comments

This manuscript presents a novel and relevant investigation that bridges plant physiology and microbial ecology. With minor editorial polishing, it will be suitable for publication.
Minor Revisions Suggested

Language and Grammar:

While the manuscript is much improved, several sentences remain awkward or overly long. I suggest one final round of light copyediting to enhance readability. For example:

“These characteristics may relate to microbial adhesion, however, they are not clearly understood.”
→ Consider revising to: “These characteristics may influence microbial adhesion, but the mechanisms are not yet fully understood.”

Terminology Consistency:

Terms like “leaf surface traits,” “morphological features,” and “structural characteristics” appear interchangeably. Choose one or define each clearly to ensure conceptual consistency throughout.

Clarify Interpretation of Statistical Results:

In several places, relationships with statistically significant p-values are described as "weak." While this is supported by low R² values, it would be helpful to explicitly mention this rationale in-text when first referring to such results (e.g., in the Results or Discussion sections), so readers understand the interpretation.

Minor Formatting Suggestions:

Check all units (e.g., mm, μm, %, CFU/g) for consistent formatting and spacing.

Ensure all tables and figure captions are self-explanatory and include the sample size (n) where appropriate.

Reference Update (optional):

Consider citing any recent literature (post-2020) on LAB colonization of plant surfaces or silage fermentation microbiomes to further strengthen the relevance of your findings.

---

## Round 0.3 · Minor Revisions

Thanks for addressing the reviewers' comments, plaese also attend the following:

1) Describe the method and algorithm used for the clustering analysis in the Methods.

2) Regarding the Abstract, Lines 25-33 do not seem to relate to the study objectives or the title of the manuscript. The abstract would have stronger impact if it were to focus on the key objectives of this study (i.e. relating bacteria to leaf measurements, rather than comparing leaf measurements among plants."

Reviewer 3 ·

Basic reporting

No comment

Experimental design

No comment

Validity of the findings

No comment

Additional comments

No comment

---

## Round 0.4 · accepted · Accept

Thanks for addressing all comments!